# Detection of High Level of Co-Infection and the Emergence of Novel SARS CoV-2 Delta-Omicron and Omicron-Omicron Recombinants in the Epidemiological Surveillance of Andalusia

**DOI:** 10.3390/ijms24032419

**Published:** 2023-01-26

**Authors:** Javier Perez-Florido, Carlos S. Casimiro-Soriguer, Francisco Ortuño, Jose L. Fernandez-Rueda, Andrea Aguado, María Lara, Cristina Riazzo, Manuel A. Rodriguez-Iglesias, Pedro Camacho-Martinez, Laura Merino-Diaz, Inmaculada Pupo-Ledo, Adolfo de Salazar, Laura Viñuela, Ana Fuentes, Natalia Chueca, Federico García, Joaquín Dopazo, Jose A. Lepe

**Affiliations:** 1Computational Medicine Platform, Andalusian Public Foundation Progress and Health-FPS, 41013 Sevilla, Spain; 2Institute of Biomedicine of Seville, IBiS, University Hospital Virgen del Rocío/CSIC/University of Sevilla, 41013 Sevilla, Spain; 3Department of Computer Architecture and Computer Technology, University of Granada, 18071 Granada, Spain; 4Unit of Microbiology, University Hospital Reina Sofia, 14004 Cordoba, Spain; 5Servicio de Microbiología, Hospital Universitario Puerta del Mar, INIBICA, Universidad de Cádiz, 11003 Cádiz, Spain; 6Servicio de Microbiología, Unidad Clínica Enfermedades Infecciosas, Microbiología y Medicina Preventiva, Hospital Universitario Virgen del Rocío, 41013 Sevilla, Spain; 7Servicio de Microbiología, Hospital Universitario San Cecilio, 18016 Granada, Spain; 8Centro de Investigación Biomédica en Red en Enfermedades Infecciosas (CIBERINFEC), ISCIII, 28046 Madrid, Spain; 9Instituto de Investigación Biosanitaria, ibs.GRANADA, 18012 Granada, Spain; 10FPS/ELIXIR-ES, Fundación Progreso y Salud (FPS), CDCA, Hospital Virgen del Rocio, 41013 Sevilla, Spain

**Keywords:** SARS-CoV-2, co-infection, recombination, surveillance, epidemiology

## Abstract

Recombination is an evolutionary strategy to quickly acquire new viral properties inherited from the parental lineages. The systematic survey of the SARS-CoV-2 genome sequences of the Andalusian genomic surveillance strategy has allowed the detection of an unexpectedly high number of co-infections, which constitute the ideal scenario for the emergence of new recombinants. Whole genome sequence of SARS-CoV-2 has been carried out as part of the genomic surveillance programme. Sample sources included the main hospitals in the Andalusia region. In addition to the increase of co-infections and known recombinants, three novel SARS-CoV-2 delta-omicron and omicron-omicron recombinant variants with two break points have been detected. Our observations document an epidemiological scenario in which co-infection and recombination are detected more frequently. Finally, we describe a family case in which co-infection is followed by the detection of a recombinant made from the two co-infecting variants. This increased number of recombinants raises the risk of emergence of recombinant variants with increased transmissibility and pathogenicity.

## 1. Introduction

Recombination has been described in coronaviruses a long time ago [1,2], and the new emerging SARS-CoV-2 is not an exception [3]. Its recombinant origin has recently been demonstrated [4]. Homologous recombination generally occurs in unsegmented positive-sense RNA viruses, such as SARS-CoV-2, by copy-choice replication when the viral RNA-polymerase complex switches from one RNA template molecule to another during replication [5]. Therefore, co-infection by distinct viruses (parental lineages of the recombinant virus) is a requirement for homologous recombination to occur. In fact, genome surveillance programs are starting to detect pervasive co-infections, even involving different SARS-CoV-2 lineages [6]. Large-scale analysis of SARS-CoV-2 sequences has also revealed that recombinant viral genomes have circulated at low levels since the beginning of the pandemic [7], with a recombination event ratio recently estimated at approximately 2.7% [8]. Inter-lineage recombination has also been described [9], including recombinants of Alpha and Delta variants of concern (VOCs) and more recently between Delta and Omicron [10,11,12]. There is an increasing concern on the possibility that such potential for recombination can finally combine in the same genome mutations that may enhance transmissibility with those that may confer immune-escape properties. Phenotypes displaying enhanced transmissibility [13] and immune-escape [14] have already been described in SARS-CoV-2. Consequently, genomic surveillance with special focus on recombinants is currently of utmost importance [3]. 

Andalusia (the southern and most populated region of Spain) implemented early in the pandemics a pilot project for genomics survey of the first-wave SARS-CoV-2 sequencing [15], which later became the Genomic surveillance circuit of Andalusia [16,17], a systematic genomic surveillance program in coordination with the Spanish Health Authority. The circuit has rendered approximately 400 SARS-CoV-2 samples sequenced per week, currently summing up over 32.000 genomes. Viral samples came from symptomatic primary care patients. The specific vaccination status of each patient was unknown but presumably most of them were vaccinated given the almost universal coverage of the vaccination campaign in Spain. Here we report the recent observation of an unexpected increase of the number of SARS-CoV-2 with no clear lineage assignment. As we describe here, a more detailed analysis of these samples carried out between January and May 2022 showed that they really corresponded to co-infections. In parallel, an increase of the number of recombinants has also been observed, whose detailed analysis demonstrated that some of them are novel SARS-CoV-2 delta-omicron and omicron-omicron recombinant variants, yet undescribed, with two break points. Our results suggest that first generation recombinants can be producing new generations of recombinants, and also indicates that probably a higher ratio of co-infection is currently occurring. Interestingly, the analysis of the evolution of a series of infections in a family showed a scenario of co-infection followed by the emergence of a new recombinant.

## 2. Results and Discussion

### 2.1. SARS-CoV-2 Sample Selection

The systematic survey of the SARS-CoV-2 genome sequences of the Andalusian genomic surveillance circuit [16,17] documents a constant replacement of SARS-CoV-2 lineages. This trend, observed everywhere, produces continuous periods of co-circulation of different lineages. The epidemiologic map of Andalusia available from the web page of the Genomic surveillance circuit of Andalusia [16] (Figure 1) documents a remarkable cocirculation of different lineages between January and May 2022, the period covered in this study, in which 7722 SARS-CoV-2 genomes were sequenced. In particular, delta and different omicron lineages were present across the different big urban nucleus and cities of the region, which constitutes a substrate for viral recombination. Sustained co-circulation increases the possibility of co-infection events, especially in a period when many of the restrictions (wearing masks, especially in public closed spaces, social distance, and others) have been lifted. In fact, during the period studied, it was observed that a total of 90 samples used in this study with a genome coverage > 90% resulted with no lineage or recombinant clade assignment (see Appendix A for details on their origin and isolation places), which were candidates for co-infection or new recombinant variants. Illumina is used for routine sequencing and in some cases (when enough sample was available) nanopore sequencing was used to validate the finding of recombinants (See Methods).

### 2.2. Detection of Co-Infection Events

The analysis of plots of distribution of mutations allowed the detection of co-infection events, as illustrated in Figure 2, where examples of two clear cases of delta-omicron and omicron-omicron co-infections are shown. Co-infections are characterized by having the common mutations (with respect to the reference genome used, the Wuhan-Hu-1, see Methods) represented in 100% of the overlapping reads while lineage-specific mutations are supported by a complementary percentage of the reads. This analysis showed that the resulting sequences of 65 samples among the 7722 sequenced between January and May 2022 were compatible with co-infections (see Appendix A). The lineages involved in the co-infection events detected agreed with the co-circulating lineages at the times and locations where these samples were collected (Figure 1). An increasing trend of co-infection events is observed, from 0.33% samples sequenced in January which raises up to 1.63% by March, although in April and May seems to reduce again (Figure 3).

### 2.3. Detection of New Recombinants

The existence of a growing number of co-infection events suggests that the likelihood of emergence of new recombinants is growing in parallel, which actually is the observed trend. In fact, Figure 3 documents an almost exponential increase of the number of recombinants observed (already described and new ones) which ranges from none observed in January, to 0.06% and 0.18% sequenced in February and March, respectively, arriving to 1.03% sequenced in April, just one month later than achieving the peak of co-infections in March. Finally, in May the number of recombinants seem to reduce again. It is interesting to note how April and May show a decreasing trend in both co-infections and recombinants as the new omicron lineages take over the existing lineages (see Figure 1A). 

Apart from the observation of some recombinant variants similar to others already described (XT, XM, XQ, XR, XAE, XN), which have most likely been introduced in the region, the mutation frequency plots suggest the existence of some potentially new recombinants (Appendix A). These potential new recombination events were subjected to further analysis with 3seq [18] and sc2rf [19]. The results obtained showed that, despite their initial resemblance to other existing recombinants, the sequences contained two breakpoints instead of the unique breakpoint harbored by the canonical recombinant lineages. 

Upon the application of the strategy to detect recombination events (see Methods) to the SARS-CoV-2 genomic sequences three novel recombinants are reported here, for which both recombination detection algorithms rendered consistent results. One of them is a recombinant of delta AY.98 and omicron BA.1.17 (Figure 4). This recombinant has been detected in two different time points for the same patient (AND21237 in 2022/03/02 and AND23187 in 2022/02/18 see next section) and was also confirmed by the sequencing of the same viral isolate as AND21237 in Oxford Nanopore (AND23055). The other two novel recombinants are composed of omicron BA.2 and omicron BA.1.1 (AND21266 and AND22043 in Figure 5) which, according to sc2rf software, are identical. Since these samples were isolated in two different, unrelated individuals, this finding suggest that this recombinant circulates at an appreciable level in the Andalusian population, given that the sampling is random and surveys less than 5% of the positive samples [16]. Appendix A shows a detail of the recombinants found in the study.

### 2.4. Detection of the Emergence of a Recombinant

Co-infections constitute melting pots from which new recombinants may eventually emerge. The Genomic surveillance circuit of Andalusia allowed the detection of a family (Appendix A) in which a co-infection event is followed by the observation of a dominant recombinant variant made from the original coinfecting variants. 

Figure 6 describes the succession of events, in which the patient (sequence ID AND23277), which was immunosuppressed due to previous organ transplantation, was infected by a delta VOC (lineage AY.98), along with the son (sequence ID AND18640) by the end of December 2021. By 2 February 2022 the patient was still infected by the lineage AY.98 (ID AND19730) and a few days later the wife showed a similar sequence (ID AND24267). However, by 9 February the patient showed a mixture of sequences corresponding to AY.98 and B.1.17, compatible with a superinfection by an omicron VOC (ID AND20071). By the end of the month this co-infection seems stable in the patient (ID AND23187), but by 2 March the co-infection, whose feature in the plot of frequency of mutations is a frequency around 50% for lineage-specific mutations and 100% for shared mutations, displays a completely different plot with 100% for all the mutations concentrated in different regions, more compatible with a recombinant (ID AND21237). The recombination was confirmed by the 3seq and sc2rf algorithms at the level of data analysis. Moreover, the sample was sequenced again with nanopore technology (See Methods), producing a new sequence (AND23055) which confirmed the sequence derived from the Illumina sequencing (AND21237). Figure 4 shows the recombination breakpoints. It is interesting to note that delta-omicron co-infections were not infrequent and a total of 7 clear cases were detected in the data presented here (see IDs AND17788, AND17799, AND19065, AND19148, AND19238, AND19468, AND19571 in Appendix A), which provide a substrate for potential recombination events like the case described here.

### 2.5. Distribution of Breakpoints along the SARS-CoV-2 Genome

The distribution of breakpoints along the SARS-CoV-2 is not random, and occur in some specific clusters. Since breakpoint determination is based on lineage-specific mutations, but most of the nucleotides are common among lineages, breakpoints are described as stretches ranging between the nearest lineage-specific mutations of both recombinant lineages.

Figure 7 depicts a histogram of such occurrences. It is interesting to note how breakpoints flank the S protein. Thus breakpoint 20,055–21,617 (with a little overlap with S, that spans from 21563 at 25384) appear at the 5′ while 25,000–25,469 (also with a little overlap with S) and 25,584–26,060 appear at the 3′. There are also other breakpoints affecting the Orf1AB.

## 3. Materials and Methods

### 3.1. Data Source

A total of 7750 SARS-CoV-2 genomes were sequenced between January and May 2022, the period covered in this study, in the sequencing facilities of Hospital San Cecilio (Granada, Spain) and the Hospital Virgen del Rocío (Sevilla, Spain) in the Genomic surveillance circuit of Andalusia. Sequences with a good sequencing quality control but no clear lineage assignment are subjected to a more detailed study described below to detect co-infections and yet undescribed recombination events.

### 3.2. SARS-CoV-2 Genome Sequencing

For short-read sequencing, RNA preparation and amplification were carried out as described in the protocols published by the ARTIC network [20] using the V3, V4, and V4.1 versions of the ARTIC primer set from Integrated DNA Technologies (Coralville, IA, USA) to create correlative amplicons covering the SARS-CoV-2 genome, after cDNA synthesis using SuperScript IV Reverse Transcriptase (Thermo Fisher Scientific, Waltham, MA, USA) 1 µL of random hexamer primers and 11 µL of extracted RNA. Libraries were prepared according to the COVID-19 ARTIC protocol (v3, v4 and v4.1, depending on the version) and Illumina DNA Prep Kit (Illumina, San Diego, CA, USA). Library quality was confirmed using the Bioanalyzer 2100 system (Agilent Technologies, Santa Clara, CA, USA). The libraries were then quantified by Qubit DNA BR (Thermo Fisher Scientific, Waltham, MA, USA), normalized, and pooled, and sequencing was performed using Illumina MiSeq, NextSeq 500/550, and NextSeq 2000 Mid Output v2.5 (300 Cycles) sequencing reagent kits.

For long-read sequencing the SARS-Cov-2 samples where sequenced with a MinION Mk1C (Oxford Nanopore, Oxford, United Kingdom) and flow cell FLO-MIN106D using the protocol “PCR tiling of SARS-CoV-2 virus with rapid barcoding and Midnight RT PCR Expansion (SQK-RBK110.96 and EXP-MRT001)”. This protocol uses an amplicon of 1200 bp [21].

### 3.3. Pipeline for Illumina Sequencing Data Processing

Illumina sequencing data were analyzed using in-house scripts and the nf-core/viralrecon pipeline software [22], version 2.4.1. Briefly, after read quality filtering, sequences for each sample are aligned to the SARS-CoV-2 isolate Wuhan-Hu-1 (GenBank accession: MN908947.3) [23] using bowtie2 algorithm [24], followed by primer sequence removal using iVar [25]. Genomic variants were identified through iVar software [26], using a minimum allele frequency threshold of 0.25 for calling variants and a filtering step to keep variants with a minimum allele frequency threshold of 0.75. Using the set of high confidence variants and the MN908947.3 genome, a consensus genome per sample was finally built using bcftools [27].

Lineage and clade assignment to each consensus genome was generated by the Pangolin [28] (pangolin v4.0.6, pangolin-data v1.8) and Nextclade [29] (Version 1.11.0) tools, respectively.

### 3.4. Pipeline for Nanopore Sequencing Data Processing

Nanopore sequencing is also used to validate recombination events in samples with enough material for a second sequencing approach. Base calling was performed in a GPU cluster with four Tesla v100 GPUs using the app guppy v.5.0.16 and the model dna_r9.4.1_450bps_hac. From high accuracy fastq produced by guppy, the quality control and consensus sequence were performed with wf-artic [30] v0.3.12 workflow. This pipeline uses Medaka [31] v1.4.3 and the model r941_min_hac_variant_g507 to polish the consensus sequences.

### 3.5. Detection of Co-Infection Events

A simple rule is followed to detect potential co-infections. A plot of the percentage of coverage of the mutations is produced using different labels for the mutations specific to lineages and another label for the common mutations. Mutations specific to lineages are retrieved from LAPIS (Lightweight API for Sequences), which uses fully public data pre-processed and hosted by Nextstrain from NCBI GenBank, as in sc2rf software [19]. When the common mutations are in the 100% range but the lineage-specific ones are in complementary proportions (e.g., 50–50, 60–40, etc.) a co-infection is highly probable. Given the simplicity of this rule, only co-infection events in which both lineages are at approximately the same ratio are easily detected. In any case, since recombination events occur by copy-choice replication when the viral RNA-polymerase complex switches between RNA templates during replication [5], these detectable co-infections are the most likely scenarios for the emergence of recombinants. 

On the other hand, when all the mutations are in the range of 100%, and lineage-specific mutations are located in non-overlapping regions, a recombinant variant is more compatible with this observation than a co-infection. In this case, specific programs are used to decide on the likelihood of the existence of a recombinant variant (see below). 

### 3.6. Detection of Recombination Events

Known recombinant variants were detected using the Nextclade [29] (Version 1.11.0) tool. In order to detect new recombination events in the SARS-CoV-2 genomic sequences, two popular applications have been used: 3seq [18] and sc2rf [19].

## 4. Conclusions

Our results show that co-infection events are occurring, a significant percentage of which have led to recombination between variants and/or lineages of SARS-CoV-2. Although these chimeric genotypes are still relatively infrequent, we have described an increase of them just after the introduction of the variant BA.2. As new mutations that influence transmissibility or threaten to limit vaccine efficacy continue to evolve and spread, the potential for recombination, that facilitate the fusion of these mutations into a single context, will continue to increase [3]. In this scenario surveillance efforts and real-time analyses to detect recombinants should be maintained to monitor the circulation and potential spread of recombinant high-fitness genotypes. Finally, we have documented that immunocompromised individuals and those around them may play a key role in the evolution of new viruses/strains.

## Figures and Tables

**Figure 1 ijms-24-02419-f001:**
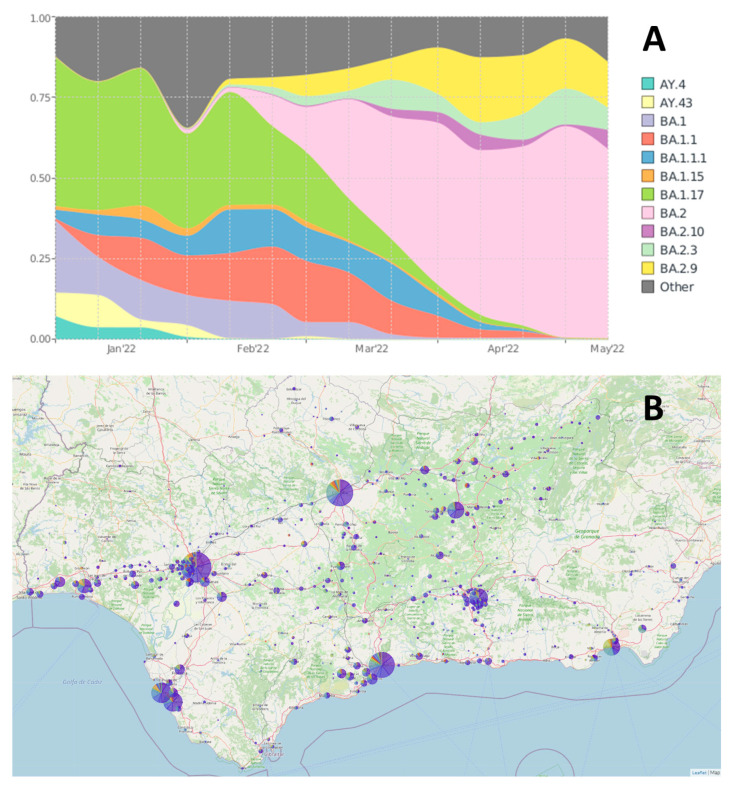
(**A**) Prevalence of main lineages in Andalusia in the period January to May 2022. (**B**) Epidemiologic map with the distribution of SARS-CoV-2 lineages in Andalusia in the same period.

**Figure 2 ijms-24-02419-f002:**
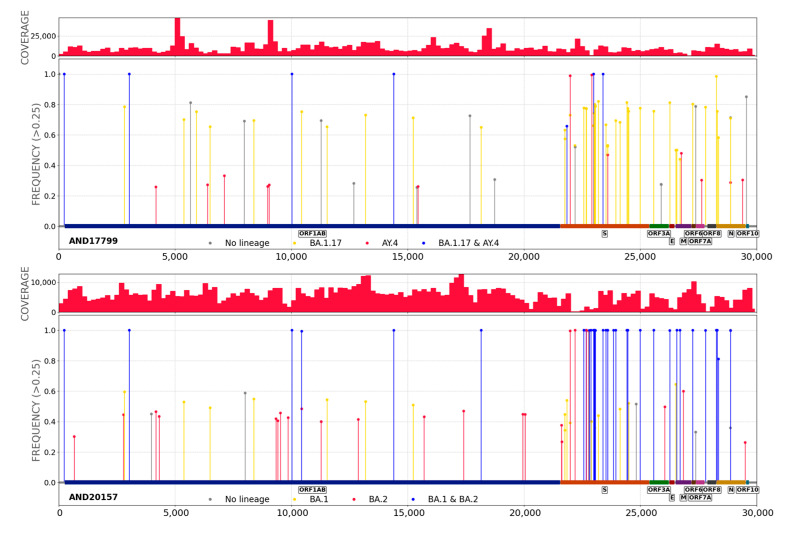
Mutation frequency plots of: (**upper panel**) delta-omicron (sample AND17799), co-infection, and (**lower panel**) omicron-omicron (sample AND20157) BA.1-BA.2 co-infection.

**Figure 3 ijms-24-02419-f003:**
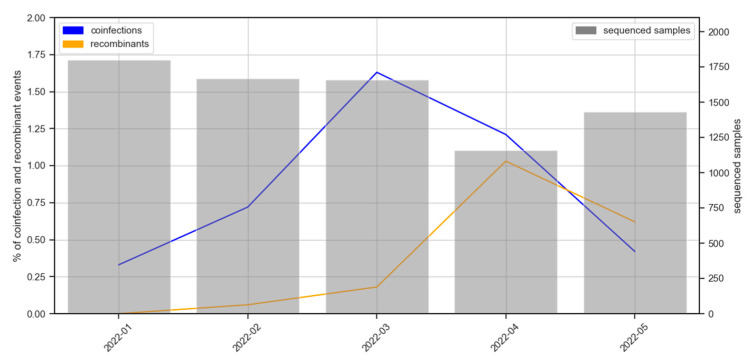
Percentage of co-infection events and recombinants observed in the samples sequenced along the period January-May 2022 (scale in the left Y axis). Bars indicate the total number of samples sequences per month (scale in the right Y axis).

**Figure 4 ijms-24-02419-f004:**
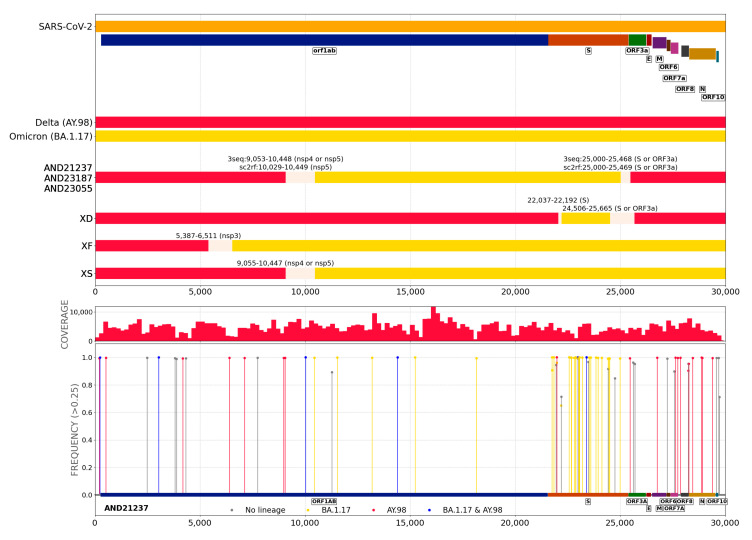
Novel Delta-omicron recombinant. Samples AND21237 and AND23187 (Illumina, San Diego, CA, USA, sequencing) and AND23055 (Oxford nanopore sequencing) correspond to the same patient (see Appendix A). 3seq outputs AY.98 and BA.1.17 lineages and their corresponding breakpoints. sc2rf outputs for Delta-BA.1 lineage with similar breakpoints to 3seq ones. A breakpoint interval (grey bar) spans between the lowest and the greatest position value of breakpoints provided by the two recombinant detection tools. Plots of distribution of mutations is also shown for AND21237 sample.

**Figure 5 ijms-24-02419-f005:**
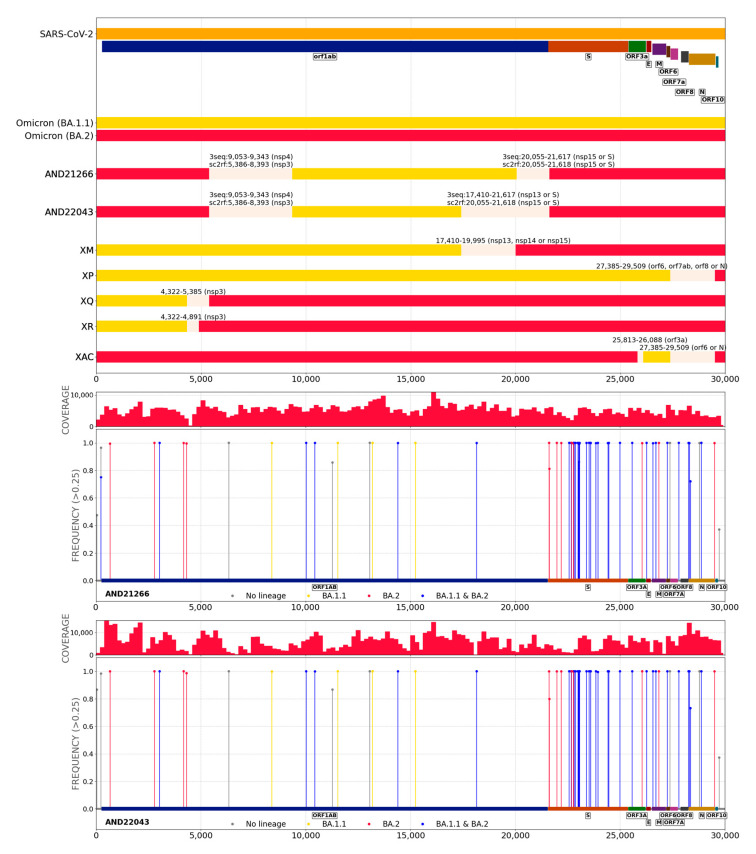
Novel omicron-omicron recombinants corresponding to two different, unrelated samples: AND21266 and AND22043 (see Appendix A). 3seq outputs for BA.2 and BA1.1 lineages and their corresponding breakpoints, sc2rf outputs BA.2 and BA.1 lineages with similar breakpoints to 3seq ones. A breakpoint interval (grey bar) spans between the lowest and the greatest position value of breakpoints provided by the two recombinant detection tools. Plots of distribution of mutations is also shown for AND21266 sample.

**Figure 6 ijms-24-02419-f006:**
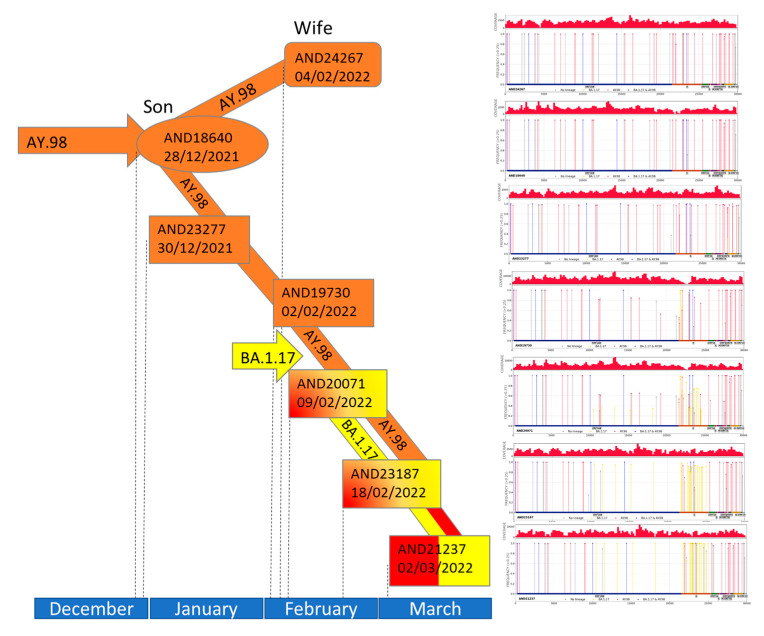
Representation in time scale of the distinct infections and co-infections by AY.98 and B.1.17 in the family in which the emergence of a recombinant was detected. The son, from which the first sequence of AY.98 was isolated is represented by an oval. The wife is represented by a rectangle with rounded corners and the rest of rectangles correspond to the patient. AY.98 is represented in red and B.1.17 in yellow. The last step between samples corresponds to the recombinant virus (arrow red and yellow). On the right part of the figure the plots of distribution of mutations corresponding to any of the samples sequenced can be found (Appendix A has a version of the plots in a larger format).

**Figure 7 ijms-24-02419-f007:**
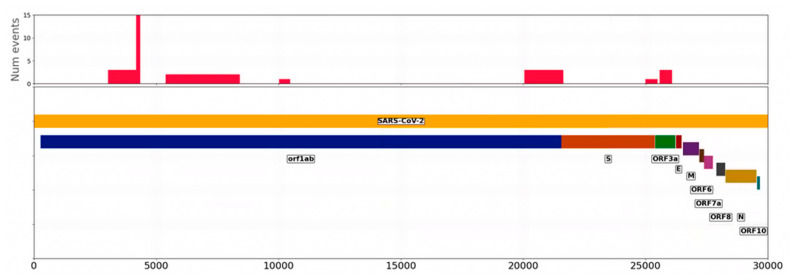
Histogram of the occurrence of breakpoints along the SARS-CoV-2 genome. The Y axis represents the number of samples in which the breakpoint was observed. The lower part represents the different SARS-CoV-2 proteins.

## Data Availability

The SARS-CoV-2 whole genome sequences described in this study are available in the European Nucleotide Archive (ENA) with the identifier PRJEB54828.

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
