# Peer review of "Detection of High Level of Co-Infection and the Emergence of Novel SARS CoV-2 Delta-Omicron and Omicron-Omicron Recombinants in the Epidemiological Surveillance of Andalusia"

_ijms, 2023, doi:10.3390/ijms24032419_

Round 1

Reviewer 1 Report

Dear authors and editorial staff:

The research article Detection of high level of co-infection and the emergence of novel SARS- CoV-2 delta-omicron and omicron-omicron recombinants in the epidemiological surveillance of Andalusia by J Perez-Florido et al. present a relatively short analysis of SARS-CoV-2 samples and sequences of interest from the Andalusian region of Spain from January – May of 2022. The methodological pipeline and consequent data were performed properly and the chosen analyses were appropriate. The figures and supplements are all clear and well-made. The English is very well written and should require a very limited amount of editing (this reviewer reads and writes English as their first language, but specifically did not comment on spelling or grammar but focused on the science). Scientifically, this paper is sound, including methods. This reviewer can recommend publication with some minor edits as noted below.

Abstract, line 34: The phrase ‘evolving towards’ should be avoided unless the authors know something about the future that the rest of us don’t. This reviewer does not agree that this article has provided evidence that co-infection or recombination is becoming more pervasive. However, the authors might be able to say something about how the changing variant landscape (Fig 1) may be correlated with changes in coinfection / superinfection (Fig 3), and therefore the potential for the emergence of recombinants.

Introduction:

-       Include a brief description of the sampling. For example – were these samples taken at hospitals, clinics, schools? Were they all symptomatic or were asymptomatic and/or close contacts to known-infected included. Were they all/mostly vaccinated? Nothing specific is needed here, but a general description is relevant.

Line 67: If this and the following few sentences are all results from this manuscript itself, then it should be mentioned here. Otherwise, if these statements are from previous publication results / analyses, please include the citations.

Line 81: “that show a more efficient transmission” – this is conjecture (and vague) unless there are studies that demonstrate this.

Line 85: Jan-May 2022 should be included in the introduction (and maybe the abstract)

Line 90: Did Andalusia specifically lift restrictions within (or immediately before) the study period of Jan-May 2022? If so, include a reference for this.

Figure 2 (and ~ line 105, ‘common mutations’): Which reference genome was used? In other words – was the Wuhan genome used as reference, thus telling the reader that the purple ‘common’ mutations are all SNVs divergent from ancestral Wuhan or maybe alpha or ?

Line 107: “65 of them” – is this 65 of the 7722 total genomes sequenced? 65 per every 1000? This should be made clear and added to the

Figure 3: The use of per 1000 (per mille, ‰) rather than percent (%) is not so common, but if the other peer reviewers are okay with ‰, this is okay. This reviewer would also like to see the average number of cases per month in the region(s) surveyed. Also important to note whether the same number of samples were assessed every week/month across Jan-May for this study.

Line 110 and throughout: Is this actually ‘per 1000 samples’? The authors should be careful in their use of ‘sequences’ and ‘genomes’ when writing about coinfections, as each (dual) coinfection is maybe 1 sample but 2+ genomes or sequences.

Line 119: The authors maybe mean to write Figure 3 here.

Section 2.3: Were all putative recombinants similarly easy to classify? In other words, is it possible that it was much easier to identify a recombinant between BA.2 and BA.1.1 or BA.2.9 (in March or April) compared to a novel recombinant between BA1.1.17 and BA.1.1 (in January) – or is this maybe the reason for the larger grey area in Figure 1?

Figure 4: This is the first time in the article that the sequencing methodologies have been mentioned. This should have come up earlier – was Illumina mostly used? Was ONT only used for the interesting samples? This, along with the library prep protocol (artic?) should be mentioned briefly either in the intro or discussion.

Samples AND21237 and AND23187 (Illumina sequencing) and AND23055 (Oxford nanopore sequencing) correspond to the same patient

2 samples from one patient on Feb 18, and then another Mar 2? Were there many long-term infections like this? This is uncommon. Was this person immune-compromised?

Line 150: repeated statement

Figure 5: The lollipop plots are identical in their unfixed SNVs, which suggests a mistake was made. The grey SNV in the middle of ORF1ab at ~87% and the red 80% SNV at the beginning of spike and the ~75% SNV at the beginning of N… etc. There is no reasonable explanation for those variants to be maintained through transmission (over a few weeks) or balanced polymorphism even if they are nonsynonymous. Further, if they are fully identical across the breakpoints, why would 3seq predict a different breakpoint range for AND22043 for the second breakpoint?

Actually – looking through the S3 figures, these apparent polymorphism variants (grey center ORF1ab, red at 80% in early spike, etc. … those are in almost all of the lollipop plots. This is either a pervasive problem or else this reviewer does not understand Chupa Chups. Either one is possible really.

Section 2.4: Oh – I see that this long-term infection makes a lot more sense now. Great that you could provide this information about the patient.

Figure 6: Not so sure if a dendrogram is the best way to present a recombination event since it necessarily should not create a tree. Perhaps a network diagram? The plots on the right are very blurry in my PDF. If possible, they should be at higher resolution in the final publication. The patient/son/wife info is kinda hidden in there too – maybe they could be box/circle/triangle or something like that?

Conclusions:

Line 262: This reviewer did not see a ‘clear upward trend’, but rather an up and down non-trend that perhaps follows new immune-evasive variant transmission, changes in human behavior and changing degrees of natural- or vaccine-derived immunity / partial-immunity.

However, this paper does not need to show that recombination happens, nor is more or less common. Perhaps most interesting points are: (1) Showing that with a new ‘wave’ (Fig 1, the BA.2s), an increase in coinfections and recombinants might be found, and (2) Through the brilliant surveillance ‘The Andalusian COVID-19 sequencing initiative’ (which should be ‘The Andalusian SARS-CoV-2 sequencing initiative’) has provided important data that has allowed researchers to use modern sequencing and analysis techniques to rapidly discover evolutionary capabilities of important viruses, (3) that immunocompromised individuals and those around them may play a key role in the evolution of new viruses / strains.

Line 205: 11 uL of swab sample in buffer?

Minor points:

The word “actually” is used a little too much throughout.

Additional results section?:

If any of the authors understand the genomics of SARS-CoV-2, would it be possible to include a small results section about the genomics of the recombinants identified in this study? For example – were the breakpoints more often in or near Spike? Did they reflect any presumptions about the waves of strains detected?

For the family case study with the immunosuppressed individual, it appears that the virus maintained the delta-derived 5’ and 3’ genome segments (from the initial delta infection) but that the eventual succeeding recombinant featuring a central omicron-derived genome (notably Spike) belonged to the second, superinfecting virus. This makes immunological sense if the host still produced some level of neutralizing antibodies, as spike is the major antigenic coding region.

Is this consistent, at least chronologically (assuming serial sampling is not usually done), for other recombinants that this study found?

Author Response

We sincerely appreciate the positive and constructive comments of the referee. The suggestions made have definitively helped to improve the manuscript. Please, find below the point-by-point responses to your questions:

QUESTION

========= 

Abstract, line 34: The phrase ‘evolving towards’ should be avoided unless the authors know something about the future that the rest of us don’t. This reviewer does not agree that this article has provided evidence that co-infection or recombination is becoming more pervasive. However, the authors might be able to say something about how the changing variant landscape (Fig 1) may be correlated with changes in coinfection / superinfection (Fig 3), and therefore the potential for the emergence of recombinants.

RESPONSE

========

We agree with the referee on this nuance. We followed his/her advice and modify the sentence accordingly.

QUESTION

========= 

Introduction:

-       Include a brief description of the sampling. For example – were these samples taken at hospitals, clinics, schools? Were they all symptomatic or were asymptomatic and/or close contacts to known-infected included. Were they all/mostly vaccinated? Nothing specific is needed here, but a general description is relevant.

 RESPONSE

========

The samples come mainly from symptomatic primary care patients. The specific vaccination status of each patient was unknown but presumably most of them were vaccinated given the almost universal coverage of the vaccination campaign in Spain. We have added a sentence in the introduction clarifying this point.

QUESTION

========= 

Line 67: If this and the following few sentences are all results from this manuscript itself, then it should be mentioned here. Otherwise, if these statements are from previous publication results / analyses, please include the citations.

RESPONSE

========

These findings are results. We have rephrased some sentences to make it clearer.

QUESTION

========= 

Line 81: “that show a more efficient transmission” – this is conjecture (and vague) unless there are studies that demonstrate this.

RESPONSE

========

The referee is right with this observation. We have substituted the sentence for another one more neutral. 

QUESTION

========= 

Line 85: Jan-May 2022 should be included in the introduction (and maybe the abstract)

RESPONSE

========

Done 

QUESTION

========= 

Line 90: Did Andalusia specifically lift restrictions within (or immediately before) the study period of Jan-May 2022? If so, include a reference for this.

RESPONSE

========

No specific change in the restrictions occurred before. We speculate that Christmas family and friends’ events may have contributed to increase the occurrence of co-infections, but we do not have any evidence for this.

QUESTION

========= 

Figure 2 (and ~ line 105, ‘common mutations’): Which reference genome was used? In other words – was the Wuhan genome used as reference, thus telling the reader that the purple ‘common’ mutations are all SNVs divergent from ancestral Wuhan or maybe alpha or ?

RESPONSE

========

The genome of reference used is the Wuhan-Hu-1, as stated in Methods. Anyway, following the comment of the referee, we have included this information for the sake of clarity.

QUESTION

========= 

Line 107: “65 of them” – is this 65 of the 7722 total genomes sequenced? 65 per every 1000? This should be made clear and added to the

RESPONSE

========

65 of the 7722. We have clarified this in the text.

QUESTION

========= 

Figure 3: The use of per 1000 (per mille, ‰) rather than percent (%) is not so common, but if the other peer reviewers are okay with ‰, this is okay. This reviewer would also like to see the average number of cases per month in the region(s) surveyed. Also important to note whether the same number of samples were assessed every week/month across Jan-May for this study.

RESPONSE

========

We have expressed the proportion of samples in %, which (we agree with the referee) is a more common way of expressing them. We have produced a new version of figure 3 in % scale and including the number of sequenced samples as requested by the referee.

QUESTION

========= 

Line 110 and throughout: Is this actually ‘per 1000 samples’? The authors should be careful in their use of ‘sequences’ and ‘genomes’ when writing about coinfections, as each (dual) coinfection is maybe 1 sample but 2+ genomes or sequences.

RESPONSE

========

This is a good point. It is actually a per sample figure measurement. We have replaced this in the text.

QUESTION

========= 

Line 119: The authors maybe mean to write Figure 3 here.

RESPONSE

========

Yes, apologies for the mistake.

QUESTION

========= 

Section 2.3: Were all putative recombinants similarly easy to classify? In other words, is it possible that it was much easier to identify a recombinant between BA.2 and BA.1.1 or BA.2.9 (in March or April) compared to a novel recombinant between BA1.1.17 and BA.1.1 (in January) – or is this maybe the reason for the larger grey area in Figure 1?

RESPONSE

========

It is expected that a combination of BA1.1.17 and BA.1.1 will be defined as a new BA.1.1* sublineage because the differences will not be very large between BA.1.1.17 and BA.1.1. It will be easier to detect recombinants between more different sublineages, such as BA.2.* and BA.1.* since the differences will be greater between them. XBB (the current one) comes to mind, which is a recombinant of BA.2.10.1 and BA.2.75. Both are BA.2.*, but it is true that BA.2.75 has always been defined as a VOC like BA.2.*. In any case, the gray area in Figure 1, is a combined contribution of unassigned and many low-proportion lineages detected.

QUESTION

========= 

Figure 4: This is the first time in the article that the sequencing methodologies have been mentioned. This should have come up earlier – was Illumina mostly used? Was ONT only used for the interesting samples? This, along with the library prep protocol (artic?) should be mentioned briefly either in the intro or discussion.

RESPONSE

========

Because of the format of the journal, the methods section, where the sequencing methodologies are described in detail, appears at the end. In any case we have added a sentence in the 2.1 subsection to introduce the sequencing technologies, as requested by the referee.

QUESTION

========= 

Samples AND21237 and AND23187 (Illumina sequencing) and AND23055 (Oxford nanopore sequencing) correspond to the same patient

2 samples from one patient on Feb 18, and then another Mar 2? Were there many long-term infections like this? This is uncommon. Was this person immune-compromised?

RESPONSE

========

Yes, as commented in more detail in section 2.4 (the referee also mentions it below)

QUESTION

========= 

Line 150: repeated statement

RESPONSE

========

Our apologies. We have removed the repeated sentence.

QUESTION

========= 

Figure 5: The lollipop plots are identical in their unfixed SNVs, which suggests a mistake was made. The grey SNV in the middle of ORF1ab at ~87% and the red 80% SNV at the beginning of spike and the ~75% SNV at the beginning of N… etc. There is no reasonable explanation for those variants to be maintained through transmission (over a few weeks) or balanced polymorphism even if they are nonsynonymous. Further, if they are fully identical across the breakpoints, why would 3seq predict a different breakpoint range for AND22043 for the second breakpoint?

Actually – looking through the S3 figures, these apparent polymorphism variants (grey center ORF1ab, red at 80% in early spike, etc. … those are in almost all of the lollipop plots. This is either a pervasive problem or else this reviewer does not understand Chupa Chups. Either one is possible really.

 RESPONSE

========

Thanks for the observations. Several things need to be clarified here. Firstly, we assume that the referee call “unfixed” SNV to those with frequencies lower than 1. If so, these “unfixed” SNVs are at similar frequencies but not equal (although in the plot look similar) Secondly, most probably these are fixed, although they are not SNVs but indels. The problem with indels is that, depending on the reading, the aligner aligns it to the left or to the right and a proportion of the readings that support a specific indel appear to support a different one. Indels are always a problem with the aligners.

Regarding the breakpoints, it is true that 3seq predicts a different breakpoint, although sc2rf predicts the same. This is because in the consensus AND22043 (which is used for the recombinant programs) there are Ns in the approximate interval [19867,20108], exactly where the breakpoint "should" be. That is why 3seq shifts it further to the left, although sc2rf does keep it exactly the same as AND21266.

QUESTION

========= 

Section 2.4: Oh – I see that this long-term infection makes a lot more sense now. Great that you could provide this information about the patient.

RESPONSE

========

Yes, this patient is a really interesting case

QUESTION

========= 

Figure 6: Not so sure if a dendrogram is the best way to present a recombination event since it necessarily should not create a tree. Perhaps a network diagram? The plots on the right are very blurry in my PDF. If possible, they should be at higher resolution in the final publication. The patient/son/wife info is kinda hidden in there too – maybe they could be box/circle/triangle or something like that?

RESPONSE

========

We have tried with this different representation based on the chain of transmission of the virus and the sequence of sampling and sequencing events. Wife, son and the patient are represented with different shapes. The plots in the right are much clearer in the figures uploaded. Probably word, or the PDF later, compresses the figure reducing the quality in the copy distributed to the referees. We have also added a new supplementary table (S5) containing full sized versions of these plots. We hope that this new representation conveys better the history.

QUESTION

========= 

Conclusions:

Line 262: This reviewer did not see a ‘clear upward trend’, but rather an up and down non-trend that perhaps follows new immune-evasive variant transmission, changes in human behavior

and changing degrees of natural- or vaccine-derived immunity / partial-immunity.

RESPONSE

========

The referee is right. We have changed the sentence suppressing this affirmation.

QUESTION

========= 

However, this paper does not need to show that recombination happens, nor is more or less common. Perhaps most interesting points are: (1) Showing that with a new ‘wave’ (Fig 1, the BA.2s), an increase in coinfections and recombinants might be found, and (2) Through the brilliant surveillance ‘The Andalusian COVID-19 sequencing initiative’ (which should be ‘The Andalusian SARS-CoV-2 sequencing initiative’) has provided important data that has allowed researchers to use modern sequencing and analysis techniques to rapidly discover evolutionary capabilities of important viruses, (3) that immunocompromised individuals and those around them may play a key role in the evolution of new viruses / strains.

 RESPONSE

========

We appreciate the synthesis made by the referee very much. We have added these ideas to the conclusions.

QUESTION

========= 

Line 205: 11 uL of swab sample in buffer?

RESPONSE

========

In fact, is RNA coming from the previous positive PCR. We have clarified this point in the text.

Minor points:

QUESTION

========= 

The word “actually” is used a little too much throughout.

 RESPONSE

========

We have reduced the number of “actually” from 8 to 4. :-)

QUESTION

========= 

Additional results section?:

If any of the authors understand the genomics of SARS-CoV-2, would it be possible to include a small results section about the genomics of the recombinants identified in this study? For example – were the breakpoints more often in or near Spike? Did they reflect any presumptions about the waves of strains detected?

RESPONSE

========

This is a very good idea. We have added a new subsection with a plot showing the distribution of breakpoints. S is flanked by two breakpoints. There are also a few breakpoints affecting Orf1AB.

QUESTION

========= 

For the family case study with the immunosuppressed individual, it appears that the virus maintained the delta-derived 5’ and 3’ genome segments (from the initial delta infection) but that the eventual succeeding recombinant featuring a central omicron-derived genome (notably Spike) belonged to the second, superinfecting virus. This makes immunological sense if the host still produced some level of neutralizing antibodies, as spike is the major antigenic coding region.

Is this consistent, at least chronologically (assuming serial sampling is not usually done), for other recombinants that this study found?

RESPONSE

========

This is a very interesting question. Unfortunately, we do not have enough samples that could be considered a chronological evolution of the viruses. Note that samples came from the different provinces of Andalusia and do not necessarily belong to the same infection event.

Reviewer 2 Report

Dear authors!

The manuscript you have prepared is well structured and, although it contains a lot of very specific information, it appears that it has been thought through to make it as easy to understand as possible.

Supplementary materials are also very well designed.

Things that could be improved - please review the conclusions section! Part of the information placed in the conclusions can be transferred to the discussion part, thus making the conclusions more specific, relevant and emphasizing the take-home message.

Author Response

We appreciate the positive comments of the referee very much and the advice for improving the manuscript.

QUESTION

========= 

Things that could be improved - please review the conclusions section! Part of the information placed in the conclusions can be transferred to the discussion part, thus making the conclusions more specific, relevant and emphasizing the take-home message.

RESPONSE

========

We appreciate very much the comment of the referee and his/her suggestion for the improvement of the manuscript. We have changed the conclusion accordingly.